# Production, Characterization and Immunomodulatory Activity of an Extracellular Polysaccharide from *Rhodotorula mucilaginosa* YL-1 Isolated from Sea Salt Field

**DOI:** 10.3390/md18120595

**Published:** 2020-11-26

**Authors:** Haifeng Li, Lifeng Huang, Yunyi Zhang, Yu Yan

**Affiliations:** 1College of Medicine, Hangzhou Normal University, Hangzhou 311121, China; lihf@hznu.edu.cn (H.L.); huang@hznu.edu.cn (L.H.); 2Department of Microbiology, Zhejiang Provincial Center for Disease Control and Prevention, Hangzhou 310051, China; 3Sargent College, Boston University, 700 Commonwealth Avenue, Boston, MA 02215, USA; yy0104@bu.edu

**Keywords:** immunomodulatory, polysaccharide, *Rhodotorula mucilaginosa* YL-1

## Abstract

A novel exopolysaccharide from marine-derived red yeast *Rhodotorula mucilaginosa* strain YL-1 was produced and characterized. The highest yield of polysaccharide reached 15.1 g/L after medium and culture parameter optimization. This exopolysaccharide, composed of four neural monosaccharides including glucose, mannose, galactose and fucose, had an average molecular weight of 1200 KDa. It had good immunomodulatory activity on RAW256.7 cell lines. ELISA (enzyme linked immunosorbent assay) and Q-PCR (quantitative real-time PCR) results showed that the cell was stimulated to express more IL-6, IL-18, IL-1β and TNFα cytokines than the control group. This is the first report of an exopolysaccharide with immunomodulatory activity from marine-derived *Rhodotorula mucilaginosa*.

## 1. Introduction

Polysaccharides from plants, microorganisms and algae have great potential value in the pharmaceutical, food and biochemical industries. Novel polysaccharides from various origins have been purified and characterized in the last few decades. Some herb plant polysaccharides exhibit anti-tumor activities [1]. Sulfate polysaccharides from marine algae possess antivirus and anti-inflammatory activity [2]. Heteropolysaccharide polysaccharides secreted by *Lactobacillus* and *Bifidobacterium* have been confirmed to possess antitumor and immunoregulation activity [3,4]. However, there are fewer reports about the biological activity of exopolysaccharides from fungi, especially marine fungi. Marine habitats yield marine microorganisms with special characteristics. The potential to discover novel polysaccharides with good biological activity from marine original fungi is of great interest. In this study, a novel extracellular polysaccharide produced by marine-derived *Rhodotorula mucilaginosa* was characterized and evaluated in terms of its immunomodulatory activity.

## 2. Results

### 2.1. Identification of Strain and Phylogenetic Tree

The YL-1 strain was isolated from water samples from a sea salt field. Its colony was light red and sticky. It was supposed to possess the capacity to produce large amounts of extracellular polysaccharides. The YL-1 strain could produce large amounts of intracellular carotene after it was grown in YPD (yeast extract peptone dextrose) medium for 2–3 days, and it existed as a light red colony on the YPD plate (Figure 1). The YL-1 strain was determined as *Rhodotorula mucilaginosa* based on 18 s rDNA sequence analysis. Phylogenetic tree results indicated that it had a close relationship with *Rhodotorula* sp. F44 (Figure 2). This yeast species is known as a producer of carotenes, oil and polysaccharides. *Rhodotorula* strains isolated from marine habits were screened for high protein content species and explored as feed additives in aquaculture.

### 2.2. Polysaccharide Production Optimization

Initial glucose of 50 g/L and yeast extract of 10 g/L were the best carbon and nitrogen sources for polysaccharide production (Figure 3a–c). Medium NaCl concentration with a range of 0–30 g/L has no obvious effect on the polysaccharide production of YL-1 (Figure 3d). The optimal temperature for YL-1 to produce polysaccharides is 24 °C (Figure 4a). When the incubation temperature was increased to above 26 °C, the production of polysaccharides decreased slightly. Oxygen was crucial for the biosynthesis of the exopolysaccharide. A rotating speed of 180 rpm was best for YL-1 polysaccharide production (Figure 4b). The maximum yield of polysaccharide of 15.1 g/L was achieved after 96 h culturation in a flask fermentation test based on the above optimized parameters (Figure 4c).

### 2.3. Purification and Structural Characterization of YL-1 Polysaccharide

#### 2.3.1. Determination of Homogeneity and Molecular Weight of Polysaccharide

The purified polysaccharide of YL-1 has two main components (ratio of 2:3) with MW of 2960 KDa and 65.7 KDa detected by analysis with a size-exclusion chromatographic column. The average molecular weight of YL-1 polysaccharide is 1200 KDa (Figure 5). It is a kind of polysaccharide with high molecular weight.

#### 2.3.2. Monosaccharide Composition Analysis

The results in Table 1 indicate that the YL-1 polysaccharide was composed of four neural monosaccharides, which were fucose, mannose, glucose and galactose, with a molar ratio of (1.9:27.6:17.6:52.9). Galactose and mannose were the main components of polysaccharide.

#### 2.3.3. Sulfate Content Determination

Sulfate groups often found in polysaccharide extracted from sea algae could provide the polysaccharide with additional biological activity. However, no sulfate content was found in the polysaccharide produced by the YL-1 strain.

#### 2.3.4. FT-IR Spectrometric Analysis

The FT-IR spectrum of YL-1 polysaccharide is shown in Figure 6: the signals at 3200, 2930, 1630, 1310 and 1000 cm^−1^ show the typical characteristics of polysaccharides. The typical broad peak at 3200 cm^−1^ was due to the stretching vibration of O-H in the sugar ring. The band around 2930 cm^−1^ was attributed to the C-H stretching vibration. The strong absorptions at around 1630 cm^−1^ and 1310 cm^−1^ indicated carboxyl groups. The peaks at 1088 cm^−1^ suggested the presence of C-H-O or C-O-C bonds in YL-1. No absorption peak at 1730 cm^−1^ was found, suggesting that YL-1 does not contain uronic acid.

#### 2.3.5. Methylation and GC-MS Analysis

The GC-MS chromatogram of YL-1 polysaccharide is shown in Figure 7. The peaks were analyzed by mass spectra and retention times to determine the glycosidic linkages of YL-1 polysaccharides. The methylation information about the YL-1 polysaccharide, shown in Table 2, indicates eight main types of sugar linkages in YL-1: Gal-(1→3), Man-(1→6), -Man-(1→2,3), -Man-(1→3,6), Glc-(1→4,6), Fuc-(1→, Glc-(1→, Man-(1→. The last tree were annotated Fuc-t, Glc-t and Man-t in GC-MS results. These results imply the main linkages of the backbone structure in YL-1. The result also indicates that galactose and mannose constituted the dominant percentage of components of YL-1, which was consistent with the monosaccharide composition analysis results. The non-reducing terminals of YL-1 contained mannose, glucose and fucose. Branches were located on mannose and contained traces of glucose residues. In addition, we could conclude from Table 2 that: (1) the content of Gal in partially methylated alditiol acetate mixtures of YL-1 accounted for approximately 52.8% of the total carbohydrate, which was consistent with the results of its sugar composition; (2) mannose residues were presented as terminal, 1,6-linked Manp, 1,2,3-linked Manp and 1,3,6-linked Manp residues; (3) galactosyl residues were presented as 1,3-linked Galp residues; (4) glucose residues were presented as terminal and 1,4,6-linked Glcp residues; (5) fucose residues were presented as terminal Fucp residues. The number of these residues (as mentioned above), except the terminal residues, accounted for 69.4% of total methylated sugars, suggesting that YL-1 was a moderate branched polysaccharide.

### 2.4. Bioactivity Test of Polysaccharide on RAW264.7 Cell Lines

#### 2.4.1. MTT Test of the Cell Viability 

To investigate the toxic effects of YL-1 polysaccharide on macrophages, the viability of cells was measured by MTT assay. As shown in Figure 8a, compared with the control group, the YL-1 polysaccharide increased the cell viability of RAW264.7 cells in the concentration range of (5–100 μg/mL). In particular, at a volume of 5 μg/mL, the polysaccharide increased the cell viability of RAW264.7 cells by 30%. The difference was significant compared with the control group (*p* < 0.01).

#### 2.4.2. NO Release Test

The level of NO in macrophages was closely related to macrophage immune activity. NO produced by activated macrophages was the main effector killing pathogenic microorganisms and tumor cells. However, excessive NO would increase the occurrence of inflammatory reactions and induce the secretion of inflammatory cytokines (e.g., TNF-α and IL-6). Lipopolysaccharide (LPS), which was an activator of macrophages, was used as a control in the tests. The level of NO of RAW267 in the 5, 50 and 200 μg/mL treatment groups increased by 1.10-, 1.85- and 3.13-fold (Figure 8b), whereas the fold change after LPS stimulation was 5.26-fold, respectively, as compared to the control group.

#### 2.4.3. Neutral Red Uptake Test

The neutral uptake activity was used to investigate the phagocytosis of macrophage cells of RAW264.7. LPS-stimulated macrophage cell has 1.31-fold activity compared with control group. The activity of neutral red uptake of RAW267 in the 12.5, 50 and 200 μg/mL polysaccharide treatment groups increased by 1.04-, 1.15- and 1.22-fold (Figure 8c). YL-1 polysaccharide has a similar effect to LPS at improving neutral uptake activity.

#### 2.4.4. Cytokine Concentration Test

To investigate the effects of YL-1 polysaccharide on IL-6,18, TNF-α and IL-1β expression of macrophages, the viability of cells was measured by ELISA Kit. For IL-1β, the minimum activated concentration of YL-1 polysaccharide was 5 μg/mL. Polysaccharide at a volume of 12.5 μg/mL has a similar activated effect to LPS at a volume of 10 μg/mL. Polysaccharide at a volume of 100 μg/mL has a better activated effect than that of LPS at a volume of 10 μg/mL (1.51-fold) (Figure 9a). As shown in Figure 9b, compared with the control group, YL-1 polysaccharide (>12.5 μg/mL) increased IL-6 expression of RAW264.7 cells, and the difference was extremely significant compared with the control group (*p* < 0.01) in the concentration range of (5–200 μg/mL). For IL-18, the minimum activated concentration of YL-1 polysaccharide was 25 μg/mL (Figure 9c). However, LPS at a volume of 10 μg/mL has a much better activated effect than that of 200 μg/mL of YL-1 polysaccharide (1.67-fold). For TNF-α, the minimum activated concentration of YL-1 polysaccharide was 50 μg/mL (Figure 9d). However, LPS at a volume of 10 μg/mL has a slightly better activated effect than that of 200 μg/mL of YL-1 (1.10-fold) polysaccharide. 

#### 2.4.5. Q-PCR Test Result of Cytokines

Transcript levels of above four cytokine genes were determined by Q-PCR. Results indicated that the mRNA levels of cytokine IL-1β genes were dramatically improved 30–90-fold by polysaccharide treatment with a concentration range of (5–200 μg/mL) compared with the control group (Figure 10a). For IL-6, IL-18 and TNF-α, the transcript levels were improved 1.3–4.5, 50–30 and 2.1–5.0-fold after polysaccharide treatment with a concentration range of (5–200 μg/mL) (Figure 10b–d). Q-PCR results corresponded with cytokine ELISA results. It could be concluded that YL-1 polysaccharide indeed significantly induced the transcription and expression of four cytokines.

## 3. Discussion

Polysaccharides with immunomodulatory activity have been studied extensively in the past several decades. In China, many traditional herbs contain various polysaccharides. Polysaccharide immunomodulatory activity was an important pharmacological activity of herbs [1]. Comparing plant polysaccharides with extensive experimental results, microbial-derived polysaccharides have been less evaluated in the medical field so far. A great deal of evidence shows that the intestinal microbiome has a close relationship with human health. Polysaccharides produced by human intestinal microorganisms such as *Lactobacillus* and *Bifidobacterium* play an important role in their probiotic mechanisms. For example, extracellular polysaccharides benefitting from intestinal adhesion of *Lactobacillus* and polysaccharides from *Bifidobacterium* could improve the cytokine expression of intestinal epithelial cells [5]. The discovery of more novel polysaccharides with good biological activity is an interesting subject of research in terms of its value for human health. In this study, we found a novel neural polysaccharide containing four monosaccharides, namely galactose, glucose, fucose and mannose, from marine red yeast *Rhodotorula mucilaginosa* YL-1. This polysaccharide with a 5–200 μg/mL working concentration could improve the transcription and expression of four cytokines, IL-1β, IL6, IL12 and TNFα in macrophage cell lines RAW264.7. In particular, for IL-1β cytokines, its transcriptional and expression levels were improved by YL-1 polysaccharide at a volume of 5 μg/mL. However, the data from Q-PCR representing the transcriptional level and ELISA representing the expression level did not demonstrate complete correspondence. After polysaccharide treatment, the improvements in the transcriptional levels of four cytokines were all greater than those of the expression levels. The relation between mRNA and protein is not strictly linear. The reason for this can be attributed to the different regulation mechanisms (such as synthesis and degradation rates) acting on both the synthesized mRNA and the synthesized protein, which affect the amounts of the two molecules differentially.

Some reports suggest that polysaccharides exert good immunomodulation on macrophage cell lines RAW264.7. Water-soluble polysaccharide CP1 extracted from *C. peregrina*, a kind of seaweed, was composed of fucose, galactose and glucose. It could improve the transcription of cytokines IL-1β, TNF-α, IL-6, IL-10 and IL-12 in RAW264.7 at a working concentration of 50 μg/mL [6]. Another Extracellular polysaccharide from fungi, *Agaricus blazei Murill*, was composed of l-fucose, l-arabinose, d-galactose, d-xylose and d-galacturonic acid. It could improve the transcription of cytokines TNF-α in RAW264.7 at a working concentration of 10 μg/mL. Removal of the terminal l-fucosyl residues by fucosidase reduced the TNF-α cytokine-stimulating activity of the polysaccharides in a RAW 264.7 test [7]. Two kinds of fucose polysaccharide, fucoidan and ascophyllan, could inhibit inflammation and induce cytokine expression and NO release in mouse RAW 264.7 cells by various pathways [8,9]. There is a possible mechanism by which YL-1 polysaccharide yields an immunodulation effect. Fucose moieties on cell-surface glycans are critical to many cell–cell interactions and signaling processes of cellular adhesion to immune regulation [10]. Fucose receptors on the surfaces of immune cells are responsible for pathogenic bacteria and virus recognition and adhesion. Immune cell surface receptor molecules such as selectin are involved in organ-specific homing. Polysaccharides with terminal l-fucosyl residues could be recognized and adhered by immune cells. Next, cell signal transduction was activated and immunodulation was started. For the YL-1 polysaccharide, it was very interesting to investigate the mechanism of structure, terminal l-fucosyl residues and immunodulation process, which will be further evaluated in our future research. On the other hand, extensive investigation indicated that a polysaccharide with nutritional value was digested and converted into short-chain fatty acids (SCFA) by the intestinal microbiota in vivo. The short-chain fatty acids acetate, propionate and butyrate are the most abundant organic anions in the human colon. SCFAs play a pivotal role in maintaining homeostasis in the colon. The possible prebiotic mechanism of SCFAs is that they can induce cell differentiation and regulate the growth and proliferation of colonic mucosal epithelial cells, whereas they reduce the growth rate of colorectal cancer cells. Colon homeostasis is vital for global human health. The nutrition mechanisms of polysaccharides were attributed to SCFAs, which can influence the human immune system and mitigate the risk of disease development [11]. It was possible that the YL-1 polysaccharide would benefit host health in vivo by modulating the host immune system using SCFA. The maximum yield of YL-1 polysaccharide of 15.1 g/L in the flask fermentation test based on optimized parameters was affordable for large-scale production. YL-1 polysaccharides with immunomodulating activity would be good candidates for commercial food additives in the future.

## 4. Materials and Methods

### 4.1. Materials

Some sand and seawater samples were collected from the Xiamen costal sea. Potato dextrose agar (PDA) medium (Hopebio, Qingdao, China) was modified by additionally adding 1.5% NaCl and 3.5% glucose. This modified PDA medium was used to isolate and culture fungi strains in samples. Culture temperature was adjusted to 26 °C to reflect the low temperatures in the natural marine habitat. After 48–72 h of culturing fungi, single colonies isolated from the PDA plate were sub-cultured and maintained on modified PDA slants.

### 4.2. Reagents

DEAE Sepharose Fast Flow and Sephacryl S-200 HR columns were purchased from GE Healthcare Life Sciences (Beijing, China). Moreover, 1 Phenyl-3-methyl-5-pyrazolone (PMP) and monosaccharide standards (l-rhamnose, l-arabinose, d-xylose, d-mannose, d-galactose, d-glucose, d-fucose) were obtained from Sigma-Aldrich Co. (St, Louis, Mo, USA). Acetonitrile for HPLC was purchased from Merck KGaA (Darmstadt, Germany). RIPM-1640 medium, fetal bovine serum (FBS) and penicillin/streptomycin were purchased from GIBCO GRL (Carlsbad, CA, USA). Moreover, 3-(4,5-dimethylthiazol-2-yl)-2,5-diphenyltetrazolium bromide (MTT) and lipopolysaccharide (LPS) were supplied by Sigma-Aldrich Co. (Shanghai, China). The NO detection kit and neural red uptake detection kit were purchased from the Beyotime Institute of Biotechnology (Haimen, China). Mouse TNF-α, IL1β and IL-6 detecting ELISA kits were purchased from eBioscience (San Diego, CA, USA). All other chemicals and reagents were of analytical grade.

### 4.3. Production Optimization of Polysaccharides

*Rhodotorula mucilaginosa* strain YL-1 maintained on PDA slant was incubated in a 250 mL flask containing 50 mL fermentation medium. For improvement of polysaccharide yields, the effects of nitrogen source and carbon source, initial carbon source concentration, NaCl concentration, culture temperature, rotator speed and culture time of yeast fermentation were investigated in shake flasks. Peptone, yeast extract, (NH_4_)_2_SO_4_ and corn steep powder were selected as nitrogen source and dosed in a concentration of 10 g/L. Sucrose, starch, glucose and maltodextrin were selected as carbon sources and dosed in concentrations of 50 g/L. Initial carbon source concentration was adjusted in six gradients of 20, 50, 100, 150, 200, 300 g/L. NaCl concentration in culture was set in gradients of 0, 0.2, 0.5, 1.0, 1.5, 2.0,2.5,3.0 g/L. Culture temperature was adjusted as 22, 24, 26, 28, 30 °C. Rotator (Hualida, Taicang, China) speed was adjusted as 100, 120, 140, 160, 180, 200 rpm. Culture time optimization was achieved by collecting samples at intervals of 12 h.

### 4.4. Extraction and Purification of Extracellular Polysaccharides

After culturing for 96 h, the culture supernatant was separated by centrifugation at 12,000 rpm for 15 min. After adding three fold volumes of cold ethanol into supernatant, the polysaccharide was precipitated and collected by centrifugation (Hualida, Taicang, China) at 12,000 rpm for 15 min. Savage reagent was used to remove protein and purify the crude polysaccharide according to reported protocols. After dialysis in deionized water, free metal ions ertr excluded. Finally, the polysaccharide solution was lyophilized to determine weight and stored at 4 °C for the next step of the research.

### 4.5. Structure Characterization of Purified Polysaccharide

#### 4.5.1. Determination of Homogeneity and Molecular Weight

The homogeneity and weight-average molecular weight (Mw) of purified polysaccharide was determined by high performance gel permeation chromatography (HPGPC, Huamei, Taizhou, China) on an Agilent1260 HPLC system. The system was equipped with columns of TSK-Gel G6000 PWXL and TSK-Gel G4000 PWXL in series and Viscotek 270 max detection system combined with refractive index (RI, Huxi, Shanghai, China) detector. Moreover, 0.1 M NaNO_3_ was used as the mobile phase under a constant flow of 0.6 mL/min. The sample was dissolved at a concentration of 5 mg/mL, filtered and 100 μL of the solution was injected. A series of glucan samples were used as calibration standards. The retention time was used to calculate the average molecular weight. OmniSEC 5.00 software (Malvern, UK) was used for the data acquisition and analysis.

#### 4.5.2. Monosaccharide Composition Analysis

Monosaccharide composition analysis of purified polysaccharide was determined with minor modifications, according to previous reports [12]. The analysis was conducted on an Agilent 1260 Series HPLC System equipped with a diode array detector. The derivatives and standard monosaccharides were analyzed on an Eclipse XDB-C18 Column (4.6 × 250 mm, 5 μm) at 30 °C with a flow rate of 1.0 mL/min. The mobile phase consisted of 0.1 M phosphate buffer (pH 6.8) and acetonitrile (84:16, *v*/*v*). The injection volume was 20 μL and the UV spectra were recorded at 250 nm.

#### 4.5.3. Sulfate Content Determination

According to a previously reported method, K_2_SO_4_ solution of 0.6 mg/mL was used to produce a standard curve [9]. BaCl_2_ solution in gelatin and trichloroacetic acid (TCA) were also used in the test, and 10 mg purified polysaccharide was digested using 5 mL 1M HCl at 100 °C for 6 h. Absorbance value of digested mixture at 360 nm was determined to calculate sulfate content in polysaccharide.

#### 4.5.4. FT-IR Spectrometric Analysis

Purified polysaccharide was grinded with KBr powder and pressed into a 1 mm pellet. The FT-IR spectrum was recorded on a FT-IR 650 spectrometer within a frequency range of 4000–400 cm^−1^. The sample was dissolved in water (0.5 mg/mL) and analyzed by a UV-VIS spectrophotometer (UV1900, Shimadzu, Japan) within the wavelength range of 190–400 nm.

#### 4.5.5. Methylation and GC-MS Analysis

Methylation and GC-MS analysis were completed at the BoRui Carbohydrate Company (Shanghai, China). Purified polysaccharide was methylated four times using a method described previously [12]. Complete methylation was confirmed by the disappearance of hydroxyl absorption in the ATR-IR spectrum (Huxi, Shanghai, China). The partially methylated alditol acetates were analyzed by gas chromatography–mass spectrometry (GC-MS) on a Shimadzu GCMS-QP 2010 instrument with an RXI-5 SIL MS column (30 m × 0.25 mm × 0.5 mm). The temperature program was as follows: the initial column temperature was set at 120 °C for 2 min, programmed from 120 to 250 °C at 3 °C/min (held for 5 min), injection port temperature was 250 °C and detector temperature was set 250 °C/min. Helium with flow velocity of 1mL/min was used as carrier gas. Data analysis was performed by GC-MS solution™ software (https://www.ssi.shimadzu.com/products/gas-chromatography-mass-spectrometry/gcmssolution-software.html, Shimadzu, Kyoto, Japan).

### 4.6. Immunomodulatory Activity of Purified Polysaccharide

#### 4.6.1. Cell Viability

Murine macrophage cell line, RAW264.7, was purchased from the Shanghai Institute of Cell Biology (Shanghai, China) and maintained in RPMI 1640 that was supplemented with 100 U/mL penicillin, 100 U/mL streptomycin and 10% fetal bovine serum. Cells were grown at 37 °C in a humidified 5% CO_2_ incubator. After adding polysaccharide and incubation, we discarded the old medium and incubated 2 × 10^4^ cell/well with 100 μL MTT solution (0.1 mg/mL) for 1 h. The MTT solution was removed and the purple crystals were dissolved with 100 μL DMSO. The absorbances of the treated samples and control were measured at 570 nm and the cell viability ratio was calculated as follows: cell viability ratio (%) = (A_treated_/A_control_) × 100%.

#### 4.6.2. Immunomodulatory Activity of Cytokines

For cytokine test experiments, the RAW264.7 cells were diluted to a density of 5 × 10^5^ cells/well and then incubated in 96-well flat-bottom plates for 24 h. The medium was then incubated with 0, 5, 12.5, 25, 50, 100, 200 μg/mL of polysaccharide or 10 μg/mL of lipopolysaccharide (LPS) and incubated for 24 h. Moreover, 10 μg/mL of Polymyxin B was also added to each well to exclude the effect of LPS, which was used as a positive reference drug in the test. Cytokines TNF-α, IL-1β, IL-6, IL-18 in the culture supernatant were assayed with the corresponding sandwich enzyme-linked immunosorbent assay (ELISA) kits according to the manufacturer’s instructions. ELISA Kit (Thermol BMS 607HS and BMS 603-2) was used to test TNF-α and IL-6. ELISA Kit (Beyotime PI301 and PI553) was used to test IL-1β and IL-18. For NO release test, RAW 264.7 cells were treated with complete medium, the polysaccharide samples and LPS for 24 h and the supernatants of cells were collected. The concentration of NO released from cells was determined by Griess Reagent System according to the instructions for use of kit (CST 13547). In brief, 50 μL different concentrations (5, 12.5, 25, 50, 100, 200 µg/mL) of the polysaccharide samples and LPS (10 µg/mL) were added to each well in triplicate. Then, 50 μL of nitrite standard solution (100, 50, 25, 12.5, 6.25, 3.13 and 1.56 µM) was added to wells in triplicate to determinate the standard curve of nitrite. Then, 50 μL of the sulfanilamide solution was added to all wells and they were incubated for 5–10 min in the dark at room temperature. Then, 50 μL of the NED solution (N-(1-Naphthyl) ethylenediamine dihydrochloride, Sigma33461) with working solution of 0.1% (*w*/*v*) was added to all wells and the cell mixture was incubated for 5–10 min at room temperature in the dark. The absorbance of each well at 540 nm was recorded using a microplate reader (Hualida, Haimen, China) within 30 min.

For neural red uptake test, the RAW264.7 cells were diluted to a density of 5 × 10 5 cells/well and then incubated in 96-well flat-bottom plates for 24 h. The medium was then incubated with 0, 5, 12.5, 25, 50, 100, 200 μg/mL of polysaccharide or 1 μg/mL of lipopolysaccharide (LPS) and incubated for 24 h. Then, 100 μL of the neural red solution (0.075%) (Solarbio G1316) was added to all wells and was incubated for 30 min. After washing three times, 100 μL of lyase buffer (0.01% acetic acid: ethanol = 1:1(1:1)) was added into every well. The absorbance of each well at 540 nm was recorded using a microplate reader within 30 min. Q-PCR was used to test the transcriptional level of TNF-α, IL-1β, IL-6, IL-18. Sequences of primers used in test are shown in Table 3. All treatments were performed in triplicate and repeated at least once, and the results were expressed by their means ±SD (standard deviation).

## 5. Conclusions

Marine fungi, including yeast and filamentous fungi, have many biotechnology applications [13]. *Rhodotorula* belong to the group of basidiomycetous yeasts and are usually isolated from various habitats, such as plants, the ocean and the atmosphere. Like other *Rhodotorula* species, *Rhodotorula mucilaginosa* could produce a large amount of β-carotene, synthesize oil, exopolysaccharide, intracellular protein, carotenoids, et al. [14,15,16]. *R. mucilaginosa* has been often used as feed yeast in aquaculture and as a whole cell biocatalyst in bioconversion [17,18]. It was noteworthy that *R. mucilaginosa* was an extreme yeast, as some strains have the pressure tolerance and adsorption capacity of a heavy metal [19,20]. There were some reports regarding polysaccharides produced by *R. mucilaginosa*. It was found that the polysaccharide from *R. mucilaginosa* has various biological activities including antitumor, antibacterial and antivirus. *R. mucilaginosa* CICC 33,013 produced a novel polysaccharide with four monosaccharides of galactose, arabinose, glucose and mannose. It demonstrates strong inhibition of human hepato carcinoma cells HepG2 with IC50 at 1.0mg/mL. Moreover, it also has scavenging activities, which reached around 60% at a concentration of 8 mg/mL [21]. *R. mucilaginosa* UANL-001L produces a novel extracellular polysaccharide (EPS) with four monosaccharides of glucose, mannose, galactose and fucose [22]. The most abundant monosaccharide in the composition of the EPS is glucose, which constitutes 82% of the composition. This polysaccharide has better antibiofilm and antibacterial activity against *Staphylococcus aureus* and *Pseudomonas aeruginosa* than EPS from *Lactobacillus plantarum* and *Streptococcus phocae*. Moreover, it exhibited no cytotoxicity in the in vivo murine test. There are also some reports of exopolysaccharides produced by *Rhodosporidium*, another known red yeast species. An acidic polysaccharide produced by *Rhodosporidium babjevae* ATCC 90,942 exhibited better DPPH scavenging activities than those of hyaluronic acid [23]. The polysaccharide from the YL-1 strain has good immunomodulatory activity on the RAW264.7 cell line. Four cytokines, IL6, IL-1β, IL18 and TNF-α, were improved at protein level and transcriptional level by polysaccharides of various content ranges. Even though many polysaccharides with immunomodulatory activity have been found, this is the first report of an immunomodulatory exopolysaccharide produced by *R. mucilaginosa* yeast isolated from marine habitats. The polysaccharide produced by the YL-1 strain with immunomodulatory activity has potential applications in pharmaceutical, food and feed industries.

## Figures and Tables

**Figure 1 marinedrugs-18-00595-f001:**
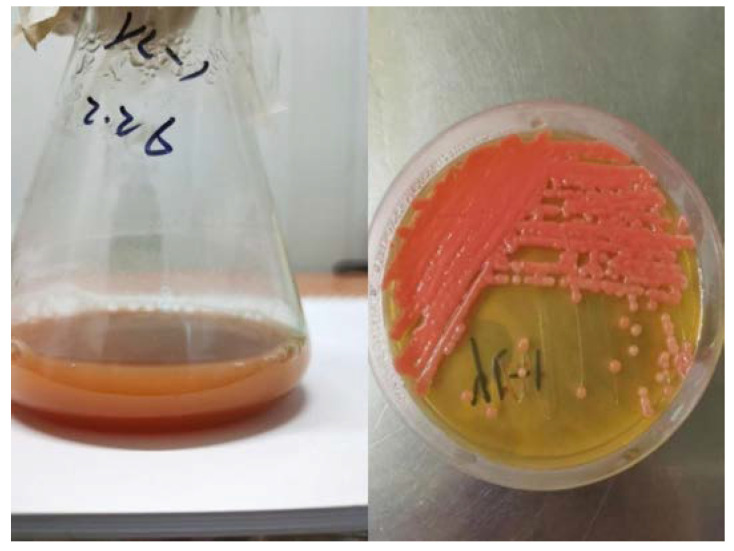
Pictures of YL-1 strain culture and colonies.

**Figure 2 marinedrugs-18-00595-f002:**
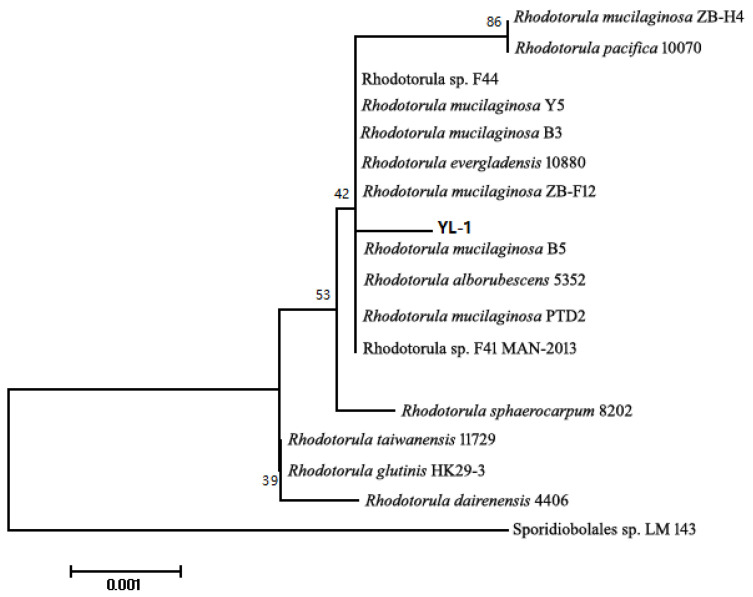
Phylogenetic tree of YL-1 strain. The phylogeny tree was computed by MEGA 7.0.14 software with the neighbor-joining statistical method. The no. of bootstrap replications was set to 1000. The analysis involved 17 nucleotide sequences. Except for YL-1 strain, the 18S rDNA sequences of other strains were obtained from NCBI Nucleotide database.

**Figure 3 marinedrugs-18-00595-f003:**
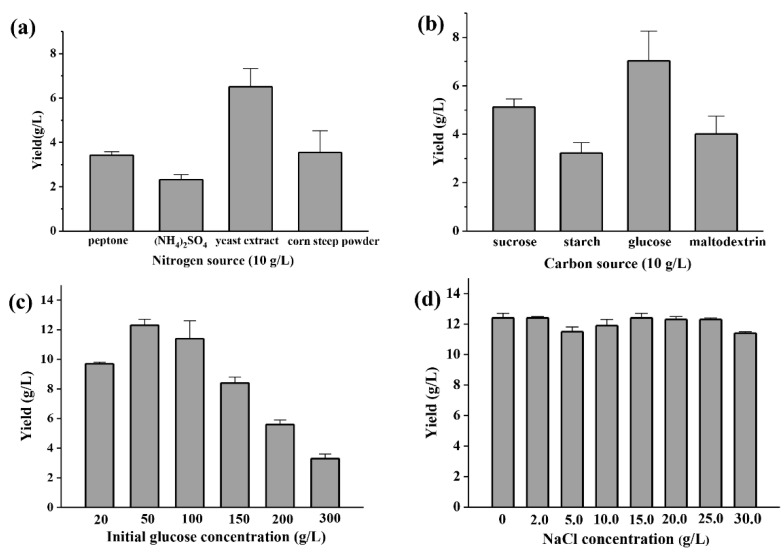
Medium optimization of YL-1 polysaccharide production. (**a**) nitrogen source; (**b**) carbon source; (**c**) initial glucose concentration; (**d**) NaCl concentration. Peptone, yeast extract, (NH4)_2_SO_4_ and corn steep powder were selected as nitrogen sources and dosed at a concentration of 10 g/L. Sucrose, starch, glucose and maltodextrin were selected as carbon sources and dosed at a concentration of 50 g/L. Initial carbon source concentration was adjusted in six gradients of 20, 50, 100, 150, 200, 300 g/L. NaCl concentration in culture was set in gradients of 0, 0.2, 0.5, 1.0, 1.5, 2.0, 2.5, 3.0 g/L. All tests were performed in triplicate and repeated at least once, and the results were expressed by their means ± SD (standard deviation). Statistical significance of the treatment effects was determined by Duncan’s multiple range *t*-test.

**Figure 4 marinedrugs-18-00595-f004:**
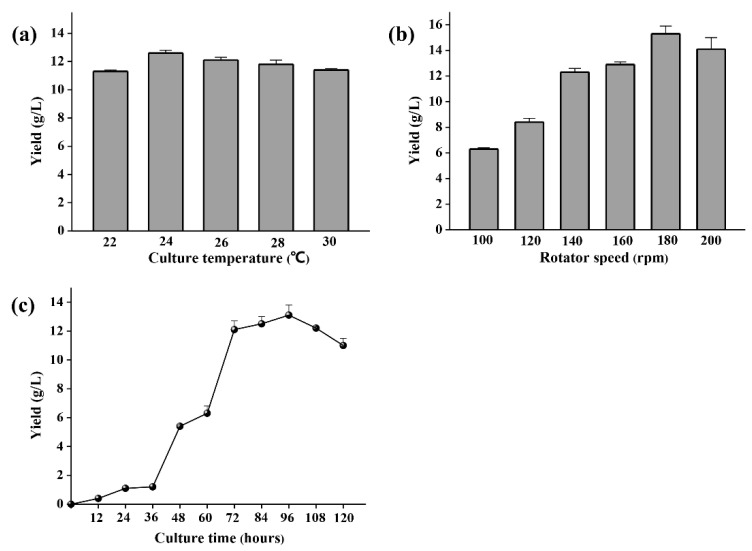
Culture parameter optimization of YL-1 polysaccharide production. (**a**) culture temperature; (**b**) rotator speed; (**c**) culture time. Culture temperature was adjusted as 22, 24, 26, 28, 30 °C. Rotator speed was adjusted as 100, 120, 140, 160, 180, 200 rpm. Culture time optimization was made by collecting samples at intervals of 12 h. All tests were performed in triplicate and repeated at least once, and the results were expressed by their means ±SD (standard deviation). Statistical significance of the treatment effects was determined by Duncan’s multiple range *t*-test.

**Figure 5 marinedrugs-18-00595-f005:**
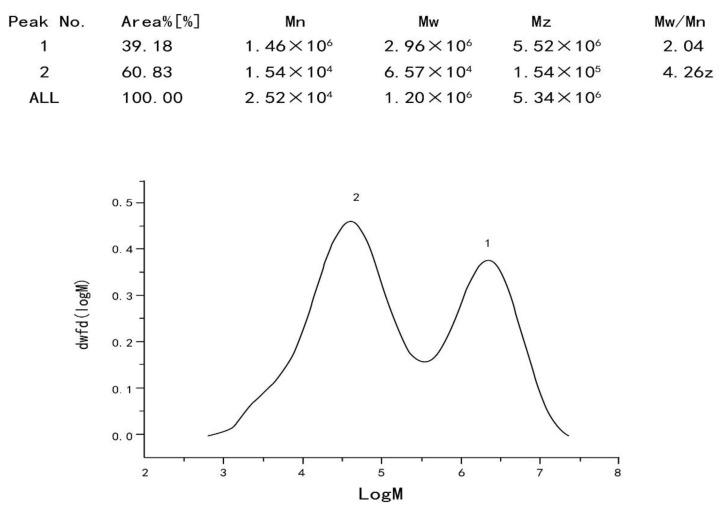
Analysis of YL-1 polysaccharide by HPGPC. The homogeneity and weight-average molecular weight (Mw) of YL-1 polysaccharide was determined by high performance gel permeation chromatography (HPGPC) on an Agilent1260 HPLC system. For the experiment, 0.1 M NaNO_3_ was used as the mobile phase under and glucan samples were used as calibration standards. The retention time was used to calculate the average molecular weight. Malvern’s OmniSEC software was used for the data acquisition and analysis.

**Figure 6 marinedrugs-18-00595-f006:**
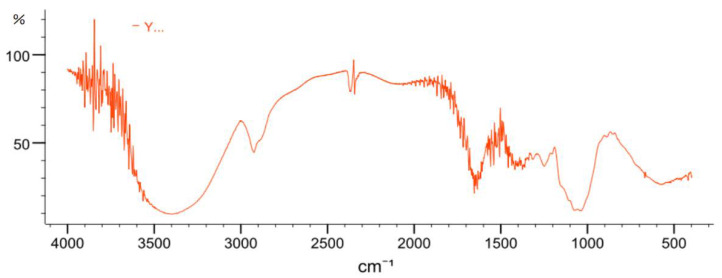
FT-IR spectrum analysis. The FT-IR spectrum was recorded on an FT-IR 650 spectrometer within the frequency range of 4000–400 cm^−1^. The sample was dissolved in water (0.5 mg/mL) and analyzed by a UV-VIS spectrophotometer (UV1900, Shimadzu, Japan) within the wavelength range of 190–400 nm.

**Figure 7 marinedrugs-18-00595-f007:**
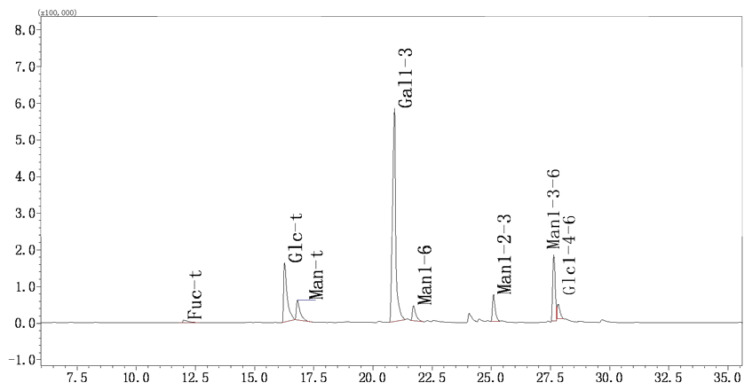
Methylation and GC-MS analysis. Complete methylation was confirmed by the disappearance of hydroxyl absorption in the ATR-IR spectrum. The partially methylated alditol acetates were analyzed by gas chromatography–mass spectrometry (GC-MS) on Shimadzu GCMS-QP 2010 instrument with an RXI-5 SIL MS column (30 m × 0.25 mm × 0.5 mm). The temperature program was as follows: the initial column temperature was set at 120 °C for 2 min, programmed from 120 to 250 °C at 3 °C/min (held for 5 min) and injection port temperature was 250 °C, while detector temperature was set 250 °C/min. Helium with flow velocity of 1 mL/min was used as carrier gas. Data analysis was performed by GC-MS solution™ software.

**Figure 8 marinedrugs-18-00595-f008:**
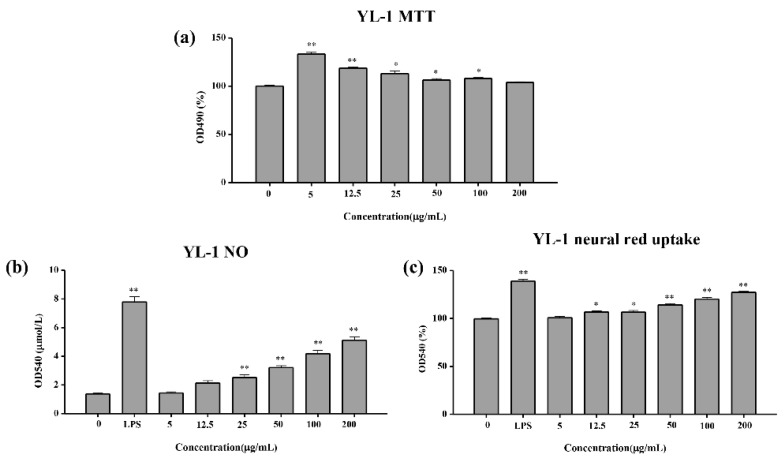
MTT, NO release and neural red uptake test results. (**a**) MTT test; (**b**) NO release test; (**c**) neural red uptake test. In MTT assay, the absorbances of the treated samples and control were measured at 570 nm and the cell viability ratio was calculated as follows: cell viability ratio (%) = (A_treated_/A_control_) × 100%. In NO assay and neural red assay, the absorbance of each well at 540 nm was recorded using a microplate reader within 30 min. All tests were performed in triplicate and repeated at least once, and the results were expressed by their means ± SD (standard deviation). Statistical significance of the treatment effects was determined by Duncan’s multiple range *t*-test. * (*p* < 0.05), ** (*p* < 0.01)

**Figure 9 marinedrugs-18-00595-f009:**
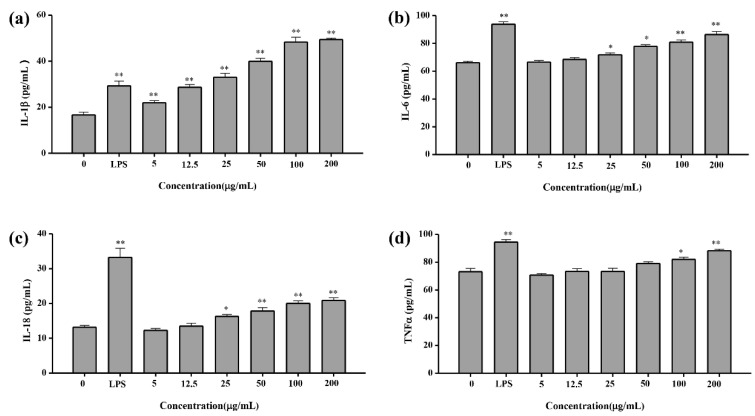
Cytokine concentration test. (**a**) IL-1β; (**b**) IL-6; (**c**) IL-18; (**d**) TNF-α. In cytokine test experiments, TNF-α, IL-1β, IL-6, IL-18 in the RAW264.7 cell culture supernatant were assayed with the corresponding sandwich enzyme-linked immunosorbent assay (ELISA). LPS was used as a positive reference drug in the test. Absorbance of each well at 450 nm was recorded using a microplate reader within 30 min. All tests were performed in triplicate and repeated at least once, and the results were expressed by their means ± SD (standard deviation). Statistical significance of the treatment effects was determined by Duncan’s multiple range *t*-test. * (*p* < 0.05), ** (*p* < 0.01)

**Figure 10 marinedrugs-18-00595-f010:**
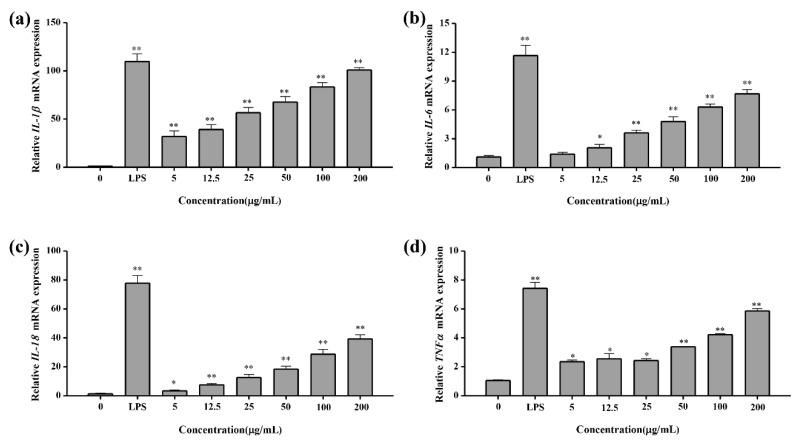
Q-PCR test of cytokines. (**a**) IL-1β; (**b**) IL-6; (**c**) IL-18; (**d**) TNF-α. All Q-PCR treatments were performed in triplicate and repeated at least once, and the results were expressed by their means ±SD (standard deviation). Statistical significance of the treatment effects was determined by Duncan’s multiple range *t*-test. * (*p* < 0.05), ** (*p* < 0.01)

**Table 1 marinedrugs-18-00595-t001:** Monosaccharide composition analysis of YL-1 polysaccharide.

Retention Time (min)	Monosaccharide Composition	Relative Molar Ratio
20.709	rhamnose	0.000
20.937	fucose	0.019
21.42	arabinose	0.000
22.075	xylose	0.000
31.52	mannose	0.276
31.763	glucose	0.176
32.562	galactose	0.529

**Table 2 marinedrugs-18-00595-t002:** Methylation analysis of polysaccharide. (Data were analyzed according to the GC–MS chromatogram (peaks) shown in Figure 7.

RT	Methylated Sugar	Mass Fragments (*m*/*z*)	Molar Ratio	Type of Linkage
12	2,3,4-Me_3_-Fucp	43, 59, 72, 89, 101, 115, 117, 131, 175	1.17	Fuc-(1→
16.26	2,3,4,6-Me_4_-Glcp	43, 71, 87, 101, 117, 129, 145, 161, 205	15	Glc-(1→
16.556	2,3,4,6-Me_4_-Manp	43, 71, 87, 101, 117, 129, 145, 161, 205	5.51	Man-(1→
22.216	2,4,6-Me3-Galp	43, 87, 99, 101, 117, 129, 161, 173, 233	52.8	→3)-Gal-(1→
22.608	2,3,4-Me3-Manp	43, 71, 87, 99, 101, 117, 129, 159, 161	3.84	→6)-Man-(1→
25.074	4,6-Me_2_-Manp	43, 85, 87, 99, 101, 127, 129, 161, 201, 216	5.36	→2,3)-Man-(1→
29.548	2,4-Me2-Manp	43, 87, 117, 129, 159, 189, 233	12.38	→3,6)-Man-(1→
27.311	2,3-Me_2_-Glcp	43, 71, 85, 87, 99, 101, 117, 127, 159, 161, 201	3.94	→4,6)-Glc-(1→

**Table 3 marinedrugs-18-00595-t003:** Primers for Q-PCR.

Gene Name	Up Primers	Down Primers
TNF-α	5-TGTCTACTGAACTTCGGGGTGAT-3	5-AACTGATGAGAGGGAGGCCAT-3
IL-1β	5-AGTTGACGGACCCCAAAAG-3	5-AGCTGGATGCTCTCATCAGG-3
IL-6	5-GCTACCAAACTGGATATAATCAGGA-3	5-CCAGGTAGCTATGGTACTCCAGAA-3
IL-18	5-GCCATGTCAGAAGACTCTTGCGTC-3	5GTACAGTGAAGTCGGCCAAAGTTGTC-3
Actin	5-GATTACTGCTCTGGCTCCTAGC-3	5-GACTCATCGTACTCCTGCTTGC-3

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
