# Peer review of "Production, Characterization and Immunomodulatory Activity of an Extracellular Polysaccharide from Rhodotorula mucilaginosa YL-1 Isolated from Sea Salt Field"

_marinedrugs, 2020, doi:10.3390/md18120595_

Round 1

Reviewer 1 Report

With interest, I read the manuscript marinedrugs-986973. I find the data presented in this work stimulating, thus, I have several suggestions. I also have several other remarks.

Comments (no special order):

  1. The manuscript needs to be deeply corrected in terms of the English language.
  2. Figure 2. The quality of the graph needs to be improved. More importantly, it should be described in respective sections how this data was obtained. Sequencing + bioinformatics? Or only bioinformatics? What samples and/or input data (sequences) sources? Methodology?
  3. Figure 3 and the rest of the figures. It is difficult to read from those figures as the fonts are too small. More importantly, it needs to be stated in the legend how the data are presented and, if appropriate (e.g. Figure 8) how the significances were calculated. Besides, the legends should be much more detailed with the description of the content of the particular graphs.
  4. Regarding statistics, the Authors describe it in the specific part of the methodology but it looks very similar between those descriptions. I would suggest summarizing it into one additional subsection “4.7. Statistics”. And, as I wrote above, I would always add a short information how the data are presented and how significances were calculated to the figure legends, in order to make it easier for the reader to understand what is shown.
  5. Figure 5. The upper panel of the figure is a table but it looks like a very raw Excel table. Please, elaborate it. Besides, all abbreviations should be explained in the figure legend.
  6. Table 1. Units need to be provided.
  7. Figure 6. What is “y” axis? I guess time but it should be written and the units given.
  8. All abbreviations used in the manuscript should be explained when introduced for the first time.
  9. Lines 156-157. Do qPCR and ELISA data fully correspond? Please, elaborate the description and, if required, address in the Discussion.
  10. Line 173 and in several other places, especially in the following sentences. “improve”. I do not think it is the best word here. Please, reformulate here and in other respective places.
  11. Lines 176-177. What you write on the pivotal role of fucose is a speculation based mostly on the literature given below? It is not supported by your own data?
  12. In the subsequent part of the Discussion, you speculate on the potential role of pattern recognition receptors in mediating the effects of polysaccharides in the immune cells, which is based on your data. Please, elaborate and write it more mechanistically.
  13. It is not clear how Rhodotorula mucilaginosa isolates were obtained. Some info is given first in lines 194-200, while it should be made clear to the Reader from the very beginning.
  14. Conclusions section is much too long. It should be made more concise and only present conclusions. Other speculative parts should be moved to the Discussion section, which is by the way too short and would benefit from being expanded.
  15. Several times, you mention a potential role of YL-1 polysaccharides in nutrition. This would possible involve also mechanisms affecting human immune system other than pattern recognition receptors. For example, human milk contains oligosaccharides that can be converted into SCFA, which in turn can influence human immune system and the risk of disease development, e.g. allergies (DOI: 3390/nu11081721; DOI: 10.3389/fped.2018.00218; DOI: 10.2217/epi-2016-0162). Would it be possible YL-1 polysaccharides used as foods would undergo a similar conversion to SCFAs (DOI: 10.1152/physrev.2001.81.3.1031 ;doi: 10.3390/nu12061802; https://doi.org/10.3389/fimmu.2020.02141). Please, speculate thoroughly in your Discussion section.

Author Response

Thank you

Reviewer 2 Report

Authors describe characterization of an extracellular polysaccharide from R. mucilaginosa YL-1 from sea salt field and immunomodulate activity using RAW256.7 cells. The results may be interesting for the researchers of this field, but the following points should be considered.

  1. Based on the methylation analysis, partial chemical structures should be drawn, if possible.
  2. Bar graph of Fig. 9 and that of Fig. 10 must be exchanged.
  3. 10, 4.5.2: The previous reports must be cited. Brief explanation of derivatives should be described.
  4. 10, 4.5.3: The previous reported method must be cited.
  5. 10, 4.5.5: The previous reported method must be cited. The commercial company of “GC-MS solution software” should be described. Shimazu?
  6. 10, 4.6.1: Cell density/well in the MTT assay should be described.

Other: Many typing errors are found. Check again please.

evidences --> evidence

Author Response

Thank you

Round 2

Reviewer 1 Report

Most, although not all, of my comments have been addressed sufficiently. Specifically:

4.Yes and no. Only legend to Figure 10 tells how the data are shown “… results were expressed by their means ±SD (standard deviation) …”. Please, add respective descriptions to the remaining figure legends.

15.Please, expand even more, as previously suggested.

In addition:

  1. Line 388. “R.murcilaginosa” -> “R. mucilaginosa”.

Author Response

4.  Respective descriptions to the remaining figure legends have been added into revised manuscript.

15. Some sentences have been added into revised manuscript.

" The short-chain fatty acids, acetate, propionate, and butyrate, are the most abundant organic anions in the human colon. SCFAs play a pivotal role in maintaining homeostasis in the colon. The possible prebiotic mechanism of SCFAs was they can induce cell differentiation and regulates growth and proliferation of colonic mucosal epithelial cells, whereas it reduces the growth rate of colorectal cancer cell. Colon homeostasis was vital for human global healthy."

16. R.murcilaginosa” has been corrected into "R. mucilaginosa”.